# Causal Inference from Text: Unveiling Interactions between Variables

**Yuxiang Zhou**[♡] and **Yulan He**[♡♢]

[♡]King's College London, [♢]The Alan Turing Institute

{yuxiang.zhou, yulan.he}@kcl.ac.uk

## Abstract

Adjusting for latent covariates is crucial for estimating causal effects from observational textual data. Most existing methods only account for *confounding* covariates that affect both *treatment* and *outcome*, potentially leading to biased causal effects. This bias arises from insufficient consideration of *non-confounding* covariates, which are relevant only to either the treatment or the outcome. In this work, we aim to mitigate the bias by unveiling interactions between different variables to disentangle the non-confounding covariates when estimating causal effects from text. The disentangling process ensures covariates only contribute to their respective objectives, enabling independence between variables. Additionally, we impose a constraint to balance representations from the treatment group and control group to alleviate selection bias. We conduct experiments on two different treatment factors under various scenarios, and the proposed model significantly outperforms recent strong baselines. Furthermore, our thorough analysis on earnings call transcripts demonstrates that our model can effectively disentangle the variables, and further investigations into real-world scenarios provide guidance for investors to make informed decisions[1].

## 1 Introduction

Causal Inference (Holland, 1985; Pearl, 2000; Morgan and Winship, 2007; Imbens and Rubin, 2015; Hernan and Robins, 2020) aims to identify how the *treatment* variable affects the *outcome* variable. For example, to estimate the effect of *"political risk"* (treatment) faced by a company on its *"stock movement"* (outcome). Early research efforts (Abadie and Imbens, 2004; Bardone-Cone and Cass, 2006; Kurth et al., 2006; Murnane and Willett, 2010; Keele, 2015) focusing on conducting randomized control trials (RCTs) to estimate

---

[1]Our code and data are released at https://github.com/zyxnlp/DIVA.

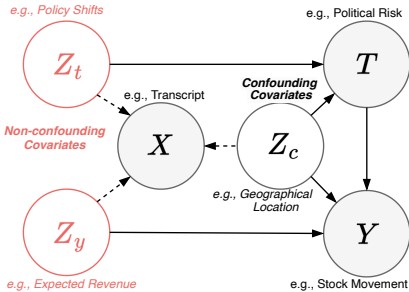

Figure 1: The causal diagram for our proposed model. Shaded nodes denote observed variables. Transparent nodes denote latent covariates derived from the transcripts, among which, nodes outlined in red represent non-confounding covariates that impact only either the treatment $T$ or outcome $O$, whereas the node outlined in black denotes the confounding covariate that influences both $T$ and $O$.

causal effects from structural numeric data have made significant progress. However, these methods requires extensive effort in treatment assignment mechanism (Halloran and Struchiner, 1995) and may suffer from ethical issues. Natural Language Processing (NLP) researchers are increasingly interested in estimating causal effects from observational unstructured text. Early literature (Choudhury et al., 2016; Olteanu et al., 2017; Pryzant et al., 2018) largely focuses on transforming texts into high-dimensional vectors using lexical features for confounding adjustment. Recent research primarily focuses on learning adequate representations through advanced NLP models. For example, Veitch et al. (2020) fine-tuned BERT (Devlin et al., 2019) to produce contextual text representations for efficient estimation of causal effects. Later, Pryzant et al. (2021) introduced strategies involving treatment enhancement and text adjustment to estimate the causal effects related to linguistic properties.

Despite their efficacy, such approaches operate under the assumption that text solely encompasses confounding covariates. This assumption raises

a potential issue due to the possible existence of unobserved non-confounding covariates that are pertain exclusively to either the treatment or the outcome. The causal estimation may be biased if we fail to differentiate non-confounding covariates from confounding ones when learning an estimation function through effective modeling of variable interactions (Pearl, 2010; Wooldridge, 2016). As illustrated in Figure 1, if we aim to accurately estimate the causal effects of treatment $T$ (e.g, Political Risk) on the outcome $Y$ (e.g., Stock Movement), we intentionally omit the consideration of the impacts originating from $Z_y$ (e.g., Expected Revenue). This mirrors our decision not to account for the influence of $Z_c$ (e.g., Geographical Location) on $Y$, as such inclusion could obfuscate our ability to discern the true effects originating from $T$.

In this paper, we propose a framework named Disentangling Interaction of VAriables (DIVA), specifically tailored for causal inference from text. We assume that the text carries sufficient information to identify the causal effects and consider the existence of non-confounding covariates. Drawing on the success of latent variable models for causal inference in literature (Louizos et al., 2017; Zhang et al., 2021), we use Variational Auto-Encoder (VAE) (Kingma and Welling, 2014) to infer confounding and non-confounding covariates. Additionally, we design a disentanglement module to ensure that covariates only contribute to their specific objectives, enabling independence between covariates. Furthermore, we propose to impose a constraint to balance representations from the treatment group and control group, which helps to mitigate selection bias.

Our contributions are summarized as follows:

- We propose the Disentangling Interaction of VAriables (DIVA) approach, tailored to mitigate the bias issue in causal inference from text.
- Our model is able to effectively model interactions among diverse variables, ensuring that each variable primarily contributes to its specific objective and promotes maximal independence.
- Our experiments demonstrate state-of-the-art results in various scenarios. A detailed analysis shows that our model effectively disentangles different variables given inherently high-dimensional nature of text representation, pro-

viding valuable insights for estimating causal effects from text.

- To the best of our knowledge, we are pioneers in addressing biased issues arising from inadequate consideration of non-confounding covariates when estimating causal effects from text.

## 2   Related Work

**Causal estimation with text data**   Early efforts in estimating causal effects from text focused on using lexical features for confounding adjustment (Choudhury et al., 2016; Choudhury and Kıcıman, 2017; Olteanu et al., 2017). Later studies investigating causal effects were devoted to effectively converting text into low-dimensional representations (Falavarjani et al., 2017; Pham and Shen, 2017; Pryzant et al., 2018; Weld et al., 2020; Cheng et al., 2021). Another line of work focused on using causal formalisms to make NLP methods more reliable (Wood-Doughty et al., 2018, 2021; Feder et al., 2021, 2022). Most recently, pre-trained language models such as BERT (Devlin et al., 2019) significantly benefited causal estimation. For example, Veitch et al. (2020) fine-tuned BERT using multi-task learning to produce contextual text representations for efficient estimation of causal effects. Later, Pryzant et al. (2021) introduced treatment-boosting and text-adjusting strategies to estimate the causal effects of linguistic properties. Our work differs from these works in three main aspects. First, we aim to mitigate the bias that arises from insufficient consideration of non-confounding covariates in causal inference. Second, we disentangle non-confounding covariates by encouraging independence among the variables, ensuring that each one contributes solely to its respective objective. Third, we introduce regularization to balance representations from the treatment group and control group, which helps to mitigate selection bias.

**Causal inference with latent variable model** Latent variable models have demonstrated their effectiveness and gained significant popularity in causal inference (Fong and Grimmer, 2016; Sridhar and Getoor, 2019; Roberts et al., 2020). For example, Louizos et al. (2017) used Variational Auto-Encoder (VAE) (Kingma and Welling, 2014) to infer confounders from latent space to estimate the effect of job training on employment following the training. Rakesh et al. (2018) inferred the causation that leads to spillover effects between

pairs of units by incorporating VAE to learn the latent attributes as confounders. We follow the line of decomposing latent factors for causal inference (Hassanpour and Greiner, 2020; Wu et al., 2020; Vowels et al., 2020; Yang et al., 2021; Zhang et al., 2021). However, there are several key distinctions in our approach. Firstly, while previous studies attempted to disentangle variables for causal inference in structured numeric data, we specifically focus on estimating causal effects from textual data. The inherently high-dimensional nature of text features presents substantial challenges in disentangling various variables within the latent space, leading to biased causal estimations. Secondly, we tailor distinct constraints to effectively model interactions among diverse variables, ensuring that each variable primarily contributes to its specific objective and promotes maximal independence. Lastly, we optimize the maximum mean discrepancy loss to achieve a balanced representation of samples from both treatment and control groups.

**NLP for earnings call transcripts** Earnings call transcripts (Frankel et al., 1997; Bowen et al., 2001; Price et al., 2011) have gained much popularity in financial analysis using NLP tools. Early work by Wang and Hua (2014) formulated financial risk prediction as a text regression task and used handcrafted features to improve SVM performance. Later, researchers (Qin and Yang, 2019; Sawhney et al., 2020; Sang and Bao, 2022; Pataci et al., 2022; Shah et al., 2022; Yang et al., 2022) focused on stock prediction by employing sophisticated neural networks with financial pragmatic features. Another line of work focused on analyzing the content of earnings call transcripts (Sawhney et al., 2021; Alhamzeh et al., 2022). For example, Keith and Stent (2019) examined analysts' decision-making behavior as it pertains to the language content of earnings calls. More in line with our work, Hassan et al. (2017) adapted linguistic tools to investigate the extent of political risk faced by firms over time and its correlation with stocks, hiring, and investment. In contrast with this prior work, our primary focus lies on estimating causal effects between financial interests, such as the impact of political risk on stocks, rather than measuring their correlations.

## 3 Preliminaries

Causal inference from text aims to estimate the causal effects based on observed textual data. Let $\mathcal{D} = \{X_i, T_i, Y_i\}_{i=1}^{N}$ represent the $N$ observational examples. Here, $X_i$ is the observed textual data (e.g., earnings call transcript) for the $i$-th example (e.g., company), and $T_i \in \{0, 1\}$ is the binary treatment variable[2]. $T_i = 1$ indicates that the $i$-th example belongs to the treatment group (e.g., a company faced high political risk). Conversely, $T_i = 0$ indicates that the $i$-th example belongs to the control group (e.g., a company faced low or no political risk). The causal effect $\tau_i$ for the $i$-th example is defined as the expected difference between its potential outcome $Y_i$ (e.g., stock volatility) of the treatment and control groups, known as the Individual Treatment Effect (ITE):

$$\tau_i = Y_i(T_i = 1) - Y_i(T_i = 0) \qquad (1)$$

One of the most challenging problems in estimating causal effects from observational data is the impossibility of simultaneously observing both potential outcomes $Y_i(T_i = 0)$ and $Y_i(T_i = 1)$ for a given example (Rubin, 1974; Holland, 1985). In other words, $\mathcal{D}$ only includes the observed outcome $Y_i$ for each example, but not the unobserved *counterfactual* outcome, which refers to the potential outcome for the $i$-th example in the alternative group. Nonetheless, it's feasible to identify the Conditional Average Treatment Effect (CATE) and the Average Treatment Effect (ATE) from observational data under certain assumptions (Spława-Neyman et al., 1990; Rubin, 1974; Pearl, 2009):

*Assumption 1* (Stable Unit Treatment Values Assumption (SUTVA)): The potential outcomes of one example are not influenced by the treatment assigned to other examples, and there are no varying forms or levels of the treatment that could result in different potential outcomes: $Y_i(t_1, ...t_i, ...t_n) = Y_i(t_i)$, and $Y(T = t_i) = Y_i(T_i)$.

*Assumption 2* (Unconfoundedness): The potential outcomes are conditionally independent of the treatment given a set of observed covariates: $(Y(1), Y(0)) \perp\!\!\!\perp T$.

*Assumption 3* (Positivity): Every individual has a non-zero probability of receiving treatment or control for all observed variables: $0 < P(T = 1|X = x) < 1$.

In line with the potential outcome framework outlined by Spława-Neyman et al. (1990) and Rubin (1974), and with the above assumptions, we

---

[2]We defer the scenarios involving multiple treatments for future exploration.

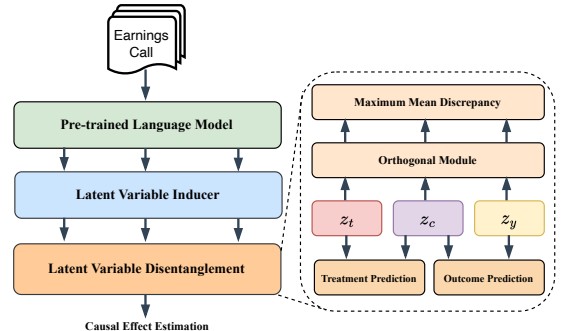

Figure 2: DIVA architecture.

can define the CATE as follows:

$$\mathbb{E}[\tau_i|X = x_i] = \mathbb{E}[Y_i(1) - Y_i(0)|X = x_i] \quad (2)$$

where $Y_i(1)$ and $Y_i(0)$ are the potential outcomes had the $i$-th individual received the treatment or control. $X$ is the observed variable which is sufficient for causal estimation. The ATE can be written as:

$$\mathbb{E}[\tau_i] = \mathbb{E}_X[\tau_i|X = x_i] \quad (3)$$

**Problem Definition** Defining $Q(t, x) = \mathbb{E}[Y_i(t)|X = x]$ as the potential outcome of observing treatment $T = t$ for an example with $X = x$, the objective is to learn an estimation function $\hat{Q}(t, x)$ that can accurately predict both the observed outcome and counterfactual outcome from $\mathcal{D}$. Therefore, we can plug in $\hat{Q}$ to estimate CATE:

$$\hat{\tau} = \frac{1}{n} \sum_{i=1}^{n} \left[ \hat{Q}(1, x_i) - \hat{Q}(0, x_i) \right] \quad (4)$$

# 4 DIVA: Disentangling Interaction of VAriables

In this section, we present the proposed Disentangling Interaction of VAriables (DIVA) framework (Figure 2) for causal inference from textual earnings call transcripts. Although previous research (Veitch et al., 2020; Pryzant et al., 2021) has explored estimating causal effects from text, one of the core contributions of our work is that we disentangle various variables to effectively model the interactions among them. This in turn enables us to learn a more accurate estimation function $\hat{Q}$ for predicting outcomes, thereby reducing the bias in the causal estimation.

Our proposed DIVA framework consists of a few steps. First, we extract the contextualized

text representation from the pre-trained language model. Following that, we employ a variational auto-encoder to determine the posterior distribution for various latent variables. Once this distribution is obtained, we use the variable disentanglement module to encourage independence among the variables, ensuring that each one contributes solely to its respective objective. Next, we utilize the disentangled variables to learn the $\hat{Q}$ function via the outcome prediction task. Finally, we plug the trained $\hat{Q}$ into a pre-determined statistic to estimate the ATE.

## 4.1 Text Encoder

Given a transcript $\boldsymbol{x} = [w_1, ..., w_n]$ that consists of $n$ words, we adopt the pre-trained language (PLM) model FinBERT (Araci, 2019)[3] to obtain the contextual representation $\boldsymbol{h}$ for each transcript:

$$\boldsymbol{h} = \mathrm{PLM}(\boldsymbol{x}) \quad (5)$$

## 4.2 Latent Variable Inducer

Inspired by recent works (Louizos et al., 2017; Zhang et al., 2021), we use the VAE to induce latent variables. Given the contextualized representation $\boldsymbol{h}$. We compute the approximation variational posterior $q_\phi(\boldsymbol{z}|\boldsymbol{h})$ using the inference network $\Phi(\boldsymbol{h}; \phi)$:

$$
\begin{aligned}
\boldsymbol{\mu} &= \boldsymbol{W}_\mu \boldsymbol{h} + \boldsymbol{b}_\mu \\
\log \boldsymbol{\sigma}^2 &= \boldsymbol{W}_\sigma \boldsymbol{h} + \boldsymbol{b}_\sigma \quad (6) \\
\boldsymbol{z} &= \boldsymbol{\mu} + \boldsymbol{\sigma} \odot \boldsymbol{\epsilon}
\end{aligned}
$$

where $\boldsymbol{W}_\mu$, $\boldsymbol{W}_\sigma$, $\boldsymbol{b}_\mu$, and $\boldsymbol{b}_\sigma$ are parameters for two MLPs. $\boldsymbol{\mu}$ and $\boldsymbol{\sigma}$ define a multivariate Gaussian distribution with a diagonal covariance matrix, and $\boldsymbol{\epsilon} \sim \mathcal{N}(0, \mathbf{I})$. Then, we sample from $q_\phi(\boldsymbol{z}|\boldsymbol{h}) \simeq \mathcal{N}(\boldsymbol{\mu}, \boldsymbol{\sigma}^2 \mathbf{I})$ to generate $\boldsymbol{z} \in \mathbb{R}^l$ as the latent representation, where $l$ is the dimension of the representation. Under the assumption that a transcript contains not only the confounding covariates, which affects both treatment and outcome, but also the non-confounding covariates specific to either the treatment or the outcome, we use separate inference networks $\Phi_c(\boldsymbol{h}; \phi_c)$ for inferring confounding covariates $\boldsymbol{z}_c$, and $\Phi_t(\boldsymbol{h}; \phi_t)$ and $\Phi_y(\boldsymbol{h}; \phi_y)$ for inferring non-confounding covariates $\boldsymbol{z}_t$ and $\boldsymbol{z}_y$, respectively. We use a one-layer parameterized MLP $\Theta(\boldsymbol{h}; \theta) := p_\theta(\boldsymbol{h}|\boldsymbol{z}_t, \boldsymbol{z}_c, \boldsymbol{z}_y)$

---

[3]We chose FinBERT due to its adaptability to text in finance domain. However, other PLMs could serve as suitable replacements.

as the decoder to reconstruct $\boldsymbol{h}$. The objective of the latent variable inducer is to maximize the evidence lower bound (ELBO):

$$\mathcal{L}_{vae} = \mathbb{E}_{\Phi_t, \Phi_c, \Phi_y}[\log \Theta(\boldsymbol{h}; \theta)] - \sum_k \text{KL}(\Phi_k || p(\boldsymbol{z}_k)) \tag{7}$$

where $k \in \{c, t, y\}$, and $p(\boldsymbol{z}_k)$ is the prior follows the Gaussian distribution $\mathcal{N}(0, \mathbf{I})$.

### 4.3 Latent Variable Disentanglement

Despite the successful application of decomposing variables in previous work (Zhang et al., 2021), unfortunately, the high-dimensional nature of text features presents significant obstacles in disentangling different variables in a latent space, leading to biased causal estimation. As will be shown in Section 5.2 (e.g., TEDVAE v.s. CEVAE), considering only non-confounding covariates, without the ability to effectively model interactions between different variables, fails to consistently achieve better performance in textual data.

To address this issue, we tailor various distinct constraints to effectively disentangle non-confounding covariates from confounding ones, ensuring that each variable primarily contributes to its specific objective and promotes maximal independence.

Specifically, we first minimize the Maximum Mean Discrepancy (MMD) (Gretton et al., 2012) loss to balance representations from the treatment group and the control group:

$$\mathcal{L}_{\text{mmd}} = \sum_{k \in \{c, t, y\}} \mathcal{M}(\boldsymbol{z}_k^{treat}; \boldsymbol{z}_k^{contl}) \tag{8}$$

where $\mathcal{M}(;)$ denotes the maximum mean discrepancy metric. $\boldsymbol{z}_k^{treat}$ and $\boldsymbol{z}_k^{contl}$ are the representations in the treatment group and the control group, respectively. The nice property of this loss is that minimizing the loss essentially reduces the discrepancy between different groups, encouraging the satisfaction of the positivity assumption. Concurrently, it promotes the inference network to generalize from the factual to counterfactual domains, leading to better counterfactual inference (Johansson et al., 2016).

Next, we introduce an orthogonal loss to maximize the independence between $\boldsymbol{z}_t$, $\boldsymbol{z}_c$, and $\boldsymbol{z}_y$ as much as possible:

$$\mathcal{L}_{\text{ort}} = \sum_{k,v} \text{Orth}(\boldsymbol{z}_k; \boldsymbol{z}_v) \tag{9}$$

where $k, v \in \{t, c, y; k \neq v\}$. $\text{Orth}(\boldsymbol{z}_k; \boldsymbol{z}_v) = ||\boldsymbol{z}_k \cdot \boldsymbol{z}_v^T - \mathbb{I}||$, and $\mathbb{I}$ is the identity matrix.

Intuitively, we expect that the prediction of the treatment label should primarily rely on $\boldsymbol{z}_t$ and $\boldsymbol{z}_c$, rather than $\boldsymbol{z}_y$. To ensure this holds, we introduce the treatment loss:

$$\mathcal{L}_t = \log P(t|\boldsymbol{z}_y) - \log P(t|\boldsymbol{z}_t, \boldsymbol{z}_c) \tag{10}$$

where $t \in \{0, 1\}$ indicates whether the transcript belongs to the treatment group.

Similarly, we expect the prediction of outcome should primarily rely on $\boldsymbol{z}_y$ and $\boldsymbol{z}_c$, and define the outcome loss:

$$\mathcal{L}_o = \mathcal{O}(y, \hat{Q}(t, \boldsymbol{z}_y, \boldsymbol{z}_c)) \tag{11}$$

where $y \in Y$ is the potential outcome. $\mathcal{O}$ is an MSE loss for real-valued outcomes and a cross-entropy loss for the binary outcomes.

The overall objective function of the latent variable disentanglement module is formulated as:

$$\mathcal{L}_d = \mathcal{L}_{vae} + \alpha \mathcal{L}_t + \beta \mathcal{L}_o + \gamma \mathcal{L}_{\text{ort}} + \eta \mathcal{L}_{\text{mmd}} \tag{12}$$

where $\alpha$, $\beta$, $\gamma$, and $\eta$ are hyper-parameters.

### 4.4 Final Training Objective

Following Veitch et al. (2020) and Pryzant et al. (2021), we introduce a Masked Language Model (MLM) objective that predicts words that are randomly[4] masked, in order to adapt text representation, making it more efficient for treatment and outcome prediction. Our final objective function is a multi-task learning objective:

$$\mathcal{L} = \mathcal{L}_d + \lambda \mathcal{L}_{\text{mlm}} \tag{13}$$

where $\lambda$ is the coefficient that balances the contribution of each component in the training process.

## 5 Experiments

We conduct experiments on both semi-synthetic data and real-world application scenarios with two objectives: 1) to empirically evaluate the effectiveness of our proposed model, and 2) to investigate practical questions in the field of finance and gain insights from the application of our model to these real-world scenarios.

---

[4] Following Devlin et al. (2019), we masked 15% of the words in each transcript.

## 5.1 Experimental Setup

**Baselines**  The baseline models selected for comparison can be broadly categorized into three groups: deep outcome regression models, latent variable models, and representation learning models. Deep outcome regression models include:

- **TARNet** Shalit et al. (2017) uses separate feed-forward networks to predict outcomes and counterfactuals.
- **CFRNet** Shalit et al. (2017) adds an integral probability metric (IPM) regularization term to TARNet to balance representation from different groups.
- **DragonNet** Shi et al. (2019) extends TARNet with an additional head adapts representation by modeling the propensity score.

The latent variable based models are:

- **CEVAE** Louizos et al. (2017) uses VAE to infer confounders from an unknown latent space to estimate causal effects.
- **TEDVAE** Zhang et al. (2021) extends CEVAE by decomposing latent factors into three sets: instrumental, confounding, and risk factors.

The representation learning models are:

- **CausalBert** Veitch et al. (2020) develops an approach to adjust for confounding features of text to estimate causal effects from observational data.
- **TextCause** Pryzant et al. (2021) introduces treatment-boosting and text-adjusting strategies to estimate causal effects of linguistic properties.

Whenever possible, we generate results for baselines using the officially released source code. In cases where the code of models is not available at the time of writing, we independently implement those models using the optimal hyper-parameter settings reported in the respective papers. For a fair comparison, we use FinBERT (Araci, 2019) to encode text for generating contextualized feature representations for all models.

**Evaluation Metric**  We evaluate the results using the precision in estimation of heterogeneous effect (PEHE) (Hill, 2011), which reflects model's individual-level estimation performance: $\sqrt{\text{PEHE}} = \sqrt{\frac{1}{N}\sum_{i=1}^{N}(\tau_i - \hat{\tau}_i)^2}$. We also report the error of ATE estimation $\delta\text{ATE} = |\tau - \frac{1}{N}\sum_{i=1}^{N}\hat{\tau}_i|$, which measure the model's population-level estimation performance.

**Setup Details**  In our experimental evaluations, each model is trained for 30 epochs with a linear warmup for the first 10% of the training steps. We employ AdamW (Loshchilov and Hutter, 2019) as the optimizer. We set the maximum learning rate at 5e-5 and use a batch size of 86. We select the optimal model weights based on either accuracy or the MSE loss of the $\hat{Q}$ function on the development set[5]. We report the average results along with the mean absolute deviations across five runs with randomly initialized parameters.

## 5.2 Experiments on Synthetic Data

### Dataset

Since ground truth causal effects ITE $\tau_i$ and ATE $\tau$, are typically inaccessible in real-world scenarios, directly training a model for causal inference is impractical. Therefore, we follow Veitch et al. (2020) and Pryzant et al. (2021), using real text and metadata to generate semi-synthetic data to empirically evaluate our proposed model. We collect 115,880 transcripts from 1,438 companies across twelve different sectors, for earnings calls held between May 2001 and October 2019. Then, we construct different datasets for two distinct treatment variables - political risk ($T_{pr}$) and sentiment ($T_s$) - under two separate scenarios: stock volatility ($Y_{vol}$) and stock movement ($Y_{mov}$). To derive $T_{pr}$, we follow Hassan et al. (2017) to calculate the political risk score[6] for each transcript. We then select the top 15,000 transcripts with the highest scores as the treatment group ($T_{pr} = 1$), indicating that the company faces high political risk. Conversely, we designate the bottom 15,000 transcripts with the lowest scores as the control group ($T_{pr} = 0$), suggesting these companies face lower or no political risk. To derive $T_s$, we follow Maia et al. (2018) and Araci (2019) to calculate the sentiment score[7] for each transcripts. We select the top 15,000 transcripts with the highest scores as the treatment group ($T_s = 1$) and select the bottom 15,000 transcripts with the lowest scores as the control group ($T_s = 0$). Finally, we simulate the outcomes by using the treatment variable $T \in \{T_{pr}, T_s\}$ along with observed covariates, $C_{size}$ and $C_{sect}$, which represent the size of the company in terms of the number of full-time employees and the industrial sector that the company operates. The real-valued stock volatility $Y_{vol}$ can

---

[5]Please refer to Appendix B for detailed hyper-parameters.
[6]https://github.com/mschwedeler/firmlevelrisk
[7]https://github.com/ProsusAI/finBERT

| Model | Political risk | | Sentiment | |
|---|---|---|---|---|
| | $\sqrt{\text{PEHE}}$ | $\delta\text{ATE}$ | $\sqrt{\text{PEHE}}$ | $\delta\text{ATE}$ |
| *Stock Volatility* | | | | |
| TARNet | 1.196±0.019 | 0.480±0.049 | 1.213±0.019 | 0.491±0.049 |
| DragonNet | 1.173±0.022 | 0.450±0.048 | 1.190±0.021 | 0.463±0.046 |
| CFRNet | 1.169±0.020 | 0.445±0.045 | 1.185±0.020 | 0.455±0.044 |
| CEVAE | 1.197±0.025 | 0.477±0.050 | 1.211±0.024 | 0.491±0.044 |
| TEDVAE | 1.212±0.056 | 0.447±0.101 | 1.228±0.056 | 0.459±0.099 |
| CausalBert | 1.097±0.032 | 0.313±0.079 | 1.121±0.034 | 0.336±0.080 |
| TextCause | 1.096±0.019 | 0.114±0.042 | 1.100±0.019 | 0.114±0.028 |
| DIVA | **1.003±0.003**[†] | **0.033±0.012**[†] | **1.010±0.007**[†] | **0.027±0.008**[†] |
| *Stock Movement* | | | | |
| TARNet | 0.497±0.001 | 0.086±0.009 | 0.497±0.001 | 0.089±0.010 |
| DragonNet | 0.497±0.003 | 0.084±0.025 | 0.497±0.004 | 0.088±0.026 |
| CFRNet | 0.497±0.003 | 0.083±0.025 | 0.497±0.004 | 0.086±0.025 |
| CEVAE | 0.499±0.004 | 0.076±0.022 | 0.499±0.004 | 0.079±0.020 |
| TEDVAE | 0.498±0.007 | 0.095±0.024 | 0.497±0.007 | 0.098±0.023 |
| CausalBert | 0.496±0.002 | 0.083±0.020 | 0.496±0.001 | 0.088±0.017 |
| TextCause | 0.526±0.008 | 0.038±0.028 | 0.522±0.009 | 0.030±0.028 |
| DIVA | **0.483±0.001**[†] | **0.009±0.004**[†] | **0.481±0.001**[†] | **0.015±0.003**[†] |

Table 1: The causal estimation results of different treatment factors on *stock volatility* and *stock movement*. Lower is better. The best results on each dataset are in bold. The second-best ones are underlined. The † marker indicates that the $p$-value is less than 0.05 compared to the second-best results. The parameter setting used is ($\alpha$=1, $\beta$=1, $\gamma$=0.5, $\epsilon$=1) for Equation (14) and (15).

be simulated as follows:

$$Y_{vol} = \alpha_v T + \beta_{v1}(\pi(C_{sect}) - \gamma_{v0}) \\ + \beta_{v2}(\pi(C_{size}) - \gamma_{v1}) + \epsilon_v \quad (14)$$

The binary stock movement (*Up* or *Down*), $Y_{mov}$ can be simulated as:

$$Y_{mov} \sim \text{Bernoulli}(\sigma(\alpha_m T + \beta_{m1}(\pi(C_{sect}) - \gamma_{m0}) \\ + \beta_{m2}(\pi(C_{size}) - \gamma_{m1}) + \epsilon_m)) \quad (15)$$

where $\pi(C_{size})$ and $\pi(C_{sect})$ are propensity socres estimated from meta data. $\alpha_v$ and $\alpha_m$ control treatment strength. $\beta_{v1}$, $\beta_{v2}$, $\beta_{m1}$, and $\beta_{m1}$ control confound strength. $\gamma_{v1}$, $\gamma_{v2}$, $\gamma_{m1}$, and $\gamma_{m2}$ are offset. $\sigma$ is the sigmoid function.

We split the dataset into the training, validation, and test sets in an 8:1:6 ratio and conduct experiments in a cross-validated manner, following Egami et al. (2018) and Pryzant et al. (2021). We conduct experiments for the two different treatment variable $T_{pr}$ and $T_s$ under the scenarios of stock volatility and stock movement, respectively. Detailed statistics of each scenario can be found in the Appendix.

**Main Results**

As shown in Table 1, DragonNet and CFRNet generally achieve better results than TARNet, suggest-

ing that additional constraints indeed benefit the outcome regression model in causal estimation. For example, DragonNet improves upon the TARNet by 0.03 in terms of $\delta$ATE based on political risk in the stock volatility scenario. We also observe that Causalbert and TextCause generally achieve better results than the deep outcome regression models such as TARNet, DragonNet, and CFRNet, as well as latent variable models such as CEVAE and TEDVAE. This suggests that the inclusion of the masked language modeling task has a positive impact on causal inference from text. Our model consistently outperforms all compared baseline models across both evaluation metrics and under both scenarios. For instance, DIVA demonstrates a significant improvement (with $p < 0.05$) over the best-performing baseline TextCause and the CausalBert model.

Interestingly, we observe that TEDVAE struggles to consistently outperform CEVAE. In particular, TEDVAE achieves better results in terms of $\delta$ATE but performs worse in terms of $\sqrt{\text{PEHE}}$ compared to CEVAE in the stock volatility scenario. We have contrary observations for TEDVAE and CEVAE under the stock movement setting. These results demonstrate that only considering non-confounding covariates, without the ability to

| Latent Covariates | | | Political Risk | | Sentiment | |
|---|---|---|---|---|---|---|
| $z_t$ | $z_c$ | $z_y$ | $\sqrt{\text{PEHE}}$ | $\delta$ATE | $\sqrt{\text{PEHE}}$ | $\delta$ATE |
| *Stock Volatility* | | | | | | |
| ✓ | | | 1.0107 | 0.0684 | 1.0179 | 0.0860 |
| ✓ | ✓ | | 1.0062 | 0.0502 | 1.0110 | 0.0519 |
| | ✓ | ✓ | 1.0054 | 0.0696 | 1.0140 | 0.0516 |
| ✓ | ✓ | ✓ | **1.0034** | **0.0332** | **1.0102** | **0.0273** |
| *Stock Movement* | | | | | | |
| ✓ | | | 0.4891 | 0.0407 | 0.4857 | 0.0358 |
| ✓ | ✓ | | 0.4900 | 0.0440 | 0.4881 | 0.0579 |
| | ✓ | ✓ | 0.4845 | 0.0390 | 0.4841 | 0.0409 |
| ✓ | ✓ | ✓ | **0.4831** | **0.0095** | **0.4814** | **0.0145** |

Table 2: Ablation study of our proposed model considering various latent covariates. Lower values are better.

effectively modeling interactions among various variables, falls short of consistently devlivering satisfactory performance in textual data. However, our DIVA model consistently surpasses both CEVAE and TEDVAE by a substantial margin across all scenarios, which clearly demonstrates the importance of the constraints we introduced and underscores the effectiveness of our proposed model to estimate causal effects more accurately from text data.

**Latent Covariates Analysis**

To further investigate the influence of various covariates on model performance, we conduct an in-depth analysis of DIVA, focusing on the disentanglement of different covariates. As shown in Table 2, merely disentangling non-confounding covariates $z_t$ or $z_y$ from the confounding covariate $z_c$ fails to consistently achieve better results compared to considering only $z_c$. Our model yields the best performance with the simultaneous disentanglement of $z_t$, $z_c$, and $z_y$. This results underscore the necessity of comprehensive covariate disentanglement, specifically, disentangling both non-confounding covariates $z_t$ and $z_y$ from the confounding covariate $z_c$, as opposed to a partial or singular focus.

**Simulation Sensitivity Analysis**

To evaluate the robustness of our proposed DIVA model, we have chosen to compare it with the two strongest baseline CausalBert and TextCause, under different simulation settings ($\alpha$=1, $\beta$=10, $\gamma$=0.5, $\epsilon$=4) in Equation (14) and (15). As shown in Table 3, our DIVA model consistently outperforms both CausalBert and TextCause across both evaluation metrics and under both scenarios. These results suggest that the superior performance of our model is not sensitive to changes in the simulation

| Model | Political Risk | | Sentiment | |
|---|---|---|---|---|
| | $\sqrt{\text{PEHE}}$ | $\delta$ATE | $\sqrt{\text{PEHE}}$ | $\delta$ATE |
| *Stock Volatility* | | | | |
| CausalBert | 4.0810 | 0.3858 | 4.1610 | 0.3904 |
| TextCause | 4.4103 | 0.2968 | 4.3927 | 0.2926 |
| DIVA | **4.0589** | **0.0534** | **4.1378** | **0.0592** |
| *Stock Movement* | | | | |
| CausalBert | 0.4992 | 0.0400 | 0.4999 | 0.0531 |
| TextCause | 0.5306 | 0.0148 | 0.5337 | 0.0286 |
| DIVA | **0.4973** | **0.0072** | **0.4966** | **0.0103** |

Table 3: Causal estimation results (lower is better) under parameter settings ($\alpha$=1, $\beta$=10, $\gamma$=0.5, $\epsilon$=4) in Equation (14) and (15).

| Model | | Political risk | | Sentiment | |
|---|---|---|---|---|---|
| | | $\sqrt{\text{PEHE}}$ | $\delta$ATE | $\sqrt{\text{PEHE}}$ | $\delta$ATE |
| *Stock Volatility* | | | | | |
| DIVA | | 1.003 | 0.033 | 1.010 | 0.027 |
| w/o | mlm | 1.004 | 0.062 | 1.011 | 0.068 |
| w/o | mmd | 1.003 | 0.040 | 1.010 | 0.032 |
| w/o | ort | 1.003 | 0.034 | 1.010 | 0.036 |
| *Stock Movement* | | | | | |
| DIVA | | 0.483 | 0.009 | 0.481 | 0.015 |
| w/o | mlm | 0.487 | 0.057 | 0.485 | 0.044 |
| w/o | mmd | 0.486 | 0.036 | 0.485 | 0.035 |
| w/o | ort | 0.486 | 0.036 | 0.485 | 0.030 |

Table 4: Ablation study of our proposed model on various scenarios. Lower values are better. 'w/o mlm' − without masked language modeling objective; 'w/o mmd' − without the Maximum Mean Discrepancy (MMD) objective; 'w/o ort' − without the orthogonal loss.

parameter setting, demonstrating the robustness or our DIVA model.

**Ablation Study**

We conducted experiments to examine the effectiveness of the major components of our proposed model. Table 4 shows the ablation results on stock volatility and stock movement scenarios. We observe that each component, namely $\mathcal{L}_{\text{mlm}}$, $\mathcal{L}_{\text{mmd}}$, and $\mathcal{L}_{\text{ort}}$ contributes to the overall performance of the model. Specifically, with the removal of the $\mathcal{L}_{\text{mmd}}$, the performance of the full model drops considerably in terms of $\sqrt{\text{PEHE}}$. Similarly, removing $\mathcal{L}_{\text{mlm}}$ results in a considerable drop in performance, measured by $\delta$ATE. These observations demonstrate the vital role played by the $\mathcal{L}_{\text{mmd}}$ regularization term, which encourages closer representations of individuals from different groups in the latent space. Incorporating the $\mathcal{L}_{\text{mlm}}$ term benefits the es-

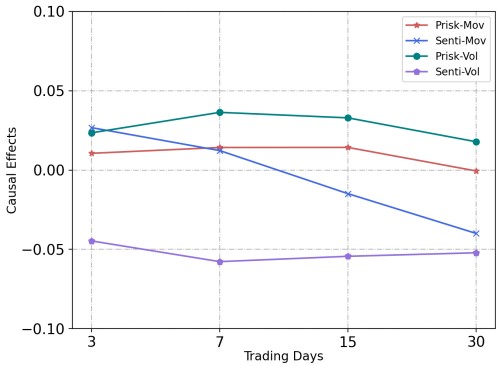

Figure 3: Causal effect of political risk and sentiment on the actual stock over trading days.

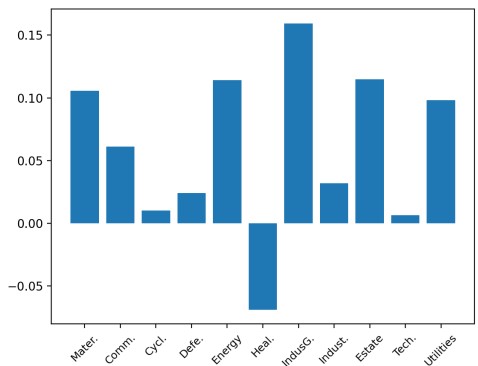

Figure 4: Causal effect of political risk on stock volatility over companies in different sectors.

timation of CATE from text data. This phenomenon aligns with previous studies such as (Veitch et al., 2020; Pryzant et al., 2018).

### 5.3 Real World Scenario Application

To answer the questions of *"How does political risk faced by a company affect its stock?"* and *"How does the sentiment conveyed in the earning call transcription of a company affect its stock?"*, we apply our proposed model to estimate the treatment effect of political risk and sentiment on actual stock volatility and stock movement.

**Stock Volatility**    Following Qin and Yang (2019) and Kogan et al. (2009), we obtain the stock prices from Yahoo Finance[8] by stock-market-scraper[9] and calculate stock volatility as:

$$v_{[t-\mu,t]} = \ln\left(\sqrt{\frac{\sum_{i=0}^{\mu}(r_{t-i} - \bar{r})^2}{\mu}}\right) \quad (16)$$

where $r_t = \frac{P_t}{P_{t-1}} - 1$ is the stock return between the close of trading day $t-1$ and day $t$, $P_t$ is the divedend-adjusted closing stock price at $t$. $\bar{r}$ is the mean of $r_t$ over the period of day $t-\mu$ to day $t$. We choose different $\mu \in \{3, 7, 15, 30\}$ to evaluate the short-term and long-term causal effects.

**Stock Movement**    Following (Medya et al., 2022), we define stock movement as:

$$m_t = \begin{cases} 1, & \text{if } r_t \geq \bar{v}_{[t-\mu,t]} \\ 0, & \text{else} \end{cases} \quad (17)$$

where $\bar{v}_{[t-\mu,t]}$ is the mean stock volatility over the period of day $t-\mu$ to day $t$.

---

[8] https://finance.yahoo.com/
[9] https://github.com/gunjannandy/stock-market-scraper

**Result**    As shown in Figure 3, we observe that the causal effects of political risk on stock increases in the short term (3 days) and begin to decline over time. Conversely, the causal effect of sentiment on stock movement decreases over time.

**Analysis**    To further investigate the effect of political risks on the stock market for different types of companies, we examine the causal effect of political risk faced by companies in different sectors on their stock prices. Figure 4 shows that the stock volatility of companies in Industrials Goods, Real Estate, and Energy are most significantly affected by the political risk they faced, while companies in Consumer Cyclical and Technology are affected to the smallest extent. The political risks faced by the Healthcare companies have no effect on their stock volatility.

## 6   Conclusion

In this paper, we propose DIVA, a novel framework designed specifically for causal inference from text. We verify its effectiveness by estimating the causal effects of treatment factor (e.g., political risk or sentiment) on a company's stock (e.g., stock volatility or movement) from the earnings conference call transcripts. The experimental results demonstrate that our model can effectively disentangle representations with different functionalities from text features by imposing constraints and utilizing multi-task learning. Furthermore, our analysis of real-world applications highlights the causal relationship between political risks faced by a company and its stock prices, providing valuable insights for the finance and investment industry.

## Limitations

Our work has a number of limitations. First, we constructed a balanced dataset in which the number of transcripts in the treatment group is equal to that in the control group. While this facilitated relatively easier causal estimation, it does not account for the selection bias that commonly exists in real-world scenarios. Consequently, causal estimation in such scenarios becomes more challenging. Second, we modeled the relation between treatment factors and stocks as a linear relation. However, in reality, this relationship is likely to be much more complex and nonlinear. A more precise modeling of this relationship would enhance the accuracy of our causal estimation.

## Acknowledgements

We would like to thank the anonymous reviewers, our meta-reviewer, and senior area chairs for their constructive comments and support with our work. We would also like to thank Rui Qiao for helpful discussions and suggestions, and Zhanming Jie for feedback on the manuscript. This work was funded by the the UK Engineering and Physical Sciences Research Council (grant no. EP/T017112/1, EP/T017112/2, EP/V048597/1). YH is supported by a Turing AI Fellowship funded by the UK Research and Innovation (grant no. EP/V020579/1, EP/V020579/2).

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

## A  Data Statistics

Table A1 shows the detailed statistics of each scenario.

| Treatment | Train | | Dev | | Test | |
|---|---|---|---|---|---|---|
| | # Treat. | # Ctrl. | # Treat. | # Ctrl. | # Treat. | # Ctrl. |
| *Stock Volatility* | | | | | | |
| Political Risk | 8,000 | 8,000 | 1,000 | 1,000 | 6,000 | 6,000 |
| Sentiment | 8,000 | 8,000 | 1,000 | 1,000 | 6,000 | 6,000 |
| *Stock Movement* | | | | | | |
| Political Risk | 8,000 | 8,000 | 1,000 | 1,000 | 6,000 | 6,000 |
| Sentiment | 8,000 | 8,000 | 1,000 | 1,000 | 6,000 | 6,000 |

Table A1: Data statistics.

## B  Hyper-parameters

Table A2 shows the detailed hyper-parameters setting of DIVA under all scenarios.

| Hyper-parameter | |
|---|---|
| Framework | Pytorch |
| GPUs | 1 A100 |
| Batch Size | 86 |
| Epoch | 30 |
| Warmup Steps | 10% |
| Learning Rate | 5.00E-05 |
| Optimizer | AdamW |
| Adam $\epsilon$ | 1E-08 |
| Max Sequence Length | 512 |
| Hidden Size | 798 |
| Hidden Layer | 12 |
| Dropout probability | 0.2 |
| Latent Dimension | 200 |
| Coefficient $\alpha$ | 1 |
| Coefficient $\beta$ | 1 |
| Coefficient $\gamma$ | 0.1 |
| Coefficient $\eta$ | 0.1 |
| Coefficient $\lambda$ | 0.01 |

Table A2: Hyper-parameters of DIVA.