# OpenReview forum: "Causal Inference from Text: Unveiling Interactions between Variables"
_EMNLP/2023/Conference — EMNLP 2023 Findings_

### Official Review · Reviewer_nCyD · 2023-07-21

**Soundness:** 3

**Excitement:**

2: Mediocre: This paper makes marginal contributions (vs non-contemporaneous work), so I would rather not see it in the conference.

**Paper Topic And Main Contributions:**

This paper studies causal effect estimation with text data, focusing on addressing the selection bias. It introduces an orthogonal module to disentangle covariates into treatment-related, outcome-related, and confounders. The disentanglement aims to ensure that covariates only contribute to the corresponding parts.

The main contributions include the design of the orthogonal module and the new dataset of 115880 transcripts.

**Questions For The Authors:**

NA

**Reasons To Accept:**

1. Understanding causal relations and estimating the effects are fundamental and challenging problems. With complex data such as text, this problem becomes even more challenging.
2. The introduced model outperforms all baselines, appearing effective.
3. The constructed datasets might help benchmark the task.

**Reasons To Reject:**

1. The studied problem of including non-confounding covariates lacks proper motivation. Using the example in Figure 1, why not include expected revenue will influence the effect of political risk on stock movement? Effect of expected revenue on stock movement is independent of the effect of political risk on stock movement. If you combine the variables of policy shifts and expected revenue, this will actually create a new confounder. It then becomes the same problem as before.
2. The proposed model is bit off to the studied problem. The problem emphasizes all non-confounding covariates, while the proposed solution targets covariates directly related to treatment- and outcome-related covariates.
3. The introduced orthogonal module is very similar to prior works such as [1], and the representation learning with VAE and representation balancing between different groups using MMD are also pretty standard. The technical contribution is limited.
4. Baselines are quite old, dating back to 2021. While showing statistical significance, some results of the proposed method are quite close to baselines, e.g., 1.096 vs. 1.003.
5. Related work should include more recent works using causal inference with text data, e.g., [2-4].


[1] Sheth, P., Guo, R., Ding, K., Cheng, L., Candan, K. S., & Liu, H. (2022, September). Causal disentanglement with network information for debiased recommendations. In International Conference on Similarity Search and Applications (pp. 265-273). Cham: Springer International Publishing.
[2] Feder, A., Keith, K. A., Manzoor, E., Pryzant, R., Sridhar, D., Wood-Doughty, Z., ... & Yang, D. (2022). Causal inference in natural language processing: Estimation, prediction, interpretation and beyond. Transactions of the Association for Computational Linguistics, 10, 1138-1158.
[3] Cheng, L., Guo, R., & Liu, H. (2022, February). Estimating causal effects of multi-aspect online reviews with multi-modal proxies. In Proceedings of the Fifteenth ACM International Conference on Web Search and Data Mining (pp. 103-112).
[4] Weld, G., West, P., Glenski, M., Arbour, D., Rossi, R. A., & Althoff, T. (2022, May). Adjusting for confounders with text: Challenges and an empirical evaluation framework for causal inference. In Proceedings of the International AAAI Conference on Web and Social Media (Vol. 16, pp. 1109-1120).

**Reproducibility:**

4: Could mostly reproduce the results, but there may be some variation because of sample variance or minor variations in their interpretation of the protocol or method.

**Reviewer Confidence:**

5: Positive that my evaluation is correct. I read the paper very carefully and I am very familiar with related work.

---

> ### Author Rebuttal · Authors · 2023-08-29
>
> Thank you for your suggestions. We hope the responses below are helpful in addressing your concerns.
>
> **R(1.1) Regarding the motivation of considering non-confounding covariates:** Our motivation, as mentioned in lines 66-85, stems from the observation that prior works in estimating causal effects from text assume text only encompasses confounding covariates. This presumption poses a potential bias issue since there are some unobserved non-confounding covariates that are only relevant to either the treatment or the outcome. We want to explore if we can mitigate the bias and accurately estimate causal effects from text, considering the existence of such types of non-confounding covariates.
>
> **R(1.2) Regarding the example in Figure 1:** We apologize for any confusion caused. The example we want to illustrated in Figure 1 is that, when focusing on accurately estimating the causal effects of “Political Risk (treatment)” on the “Stock Movement (Outcome)”, we intentionally omit the consideration of the impacts stemming from  the “Expected Revenue” This parallels our disregard for factoring in the influence of the “Geographical Location (confounder)” on the “Stock Movement (Outcome)”, as such an inclusion could obfuscate our ability to discern the true effects originating from the "Political Risk.". Our objective is to learn a more accurate estimation function $\hat Q$ in equation (4) by efficiently unveiling the interaction between these variables.
>
> We will make this part clearer in the revised version.
>
> **R(1.3) Regarding combining the variables will create a new confounder:** As highlighted in our response to  R(1.1), our rationale behind assuming the existence of non-confounding covariates exclusively relevant to either the treatment or the outcome arises from the recognition that considering only the confounding covariates in text can lead to the bias issue, which has been demonstrated by [5,6]. A similar assumption has been made in causal inference with structured numeric data such as [7] (TEDVAE baseline we compared in Table 1).
>
> As we have analyzed in lines 554-562, disentangling latent variables without imposing any constraints is inefficient. Considering only non-confounding covariates, without the ability to effectively model interactions between different variables fails to  consistently achieve better performance in textual data (e.g., TEDVAE v.s. CEVAE on the stock movement scenario). However, our proposed model consistently outperforms all compared baseline models across evaluation metrics and scenarios, which demonstrated the necessity of imposing constraints and the effectiveness of our proposed model DIVA.
>
>
> **R(2) Regarding the proposed model is a bit off to the studied problem:** Our intention is not to encompass ALL non-confounding covariates. As discussed in lines 66-85, previous attempts at estimating causal effects from text have presumed that text solely contains confounding covariates, potentially introducing bias. We want to explore if we can mitigate the bias and accurately estimate causal effects from text, considering the existence of such types of non-confounding covariates. To achieve this, we proposed using VAE to infer the unobserved non-confounding covariates and designed a disentanglement module to differentiate those non-confounding covariates from confounding ones. Additionally, we tailored various loss functions to accurately model interactions between different variables, ensuring that variables only contribute to their specific objectives and enabling independence from each other as possible.
>
> We will make this part clearer in the revised version.
>
>
> **R(3.1) Regarding orthogonal module similar to prior work [1]:** The Orthogonal Module in Figure 2 corresponds to equation (9), which was not used in [1]. We want to highlight some crucial  differences between [1] and our proposed method:
>
> 1. As illustrated in Fig 1 of [1], the authors regard the rich source of information gathered from user-item interaction and social network as a suitable proxy for accounting for confounders. They essentially assume that information only contains confounding covariates that affect both treatment and outcome. This diverges fundamentally  from our conceptualization of causal graphs when estimating causal effects (e.g., CATE) from text. Using Fig 1 of [1] as an example, if we aim to estimate causal effects of “Exposure”(treatment) on “Rating” (outcome) by conditioning only on the “rich source of information” (represented by the dashed box in Fig 1), the estimation may suffer from bias. This is because the backdoor path “exposure<-Confounders->Ratings” remains open, introducing a non-causal association. In contrast, within our assumed causal graph, all backdoor paths are blocked when estimating the causal effects (e.g, CATE) of “Political Risk” on “Stock Movement” using equation (4) upon conditioning on earnings call transcripts (Figure 1 in our paper), all of the association that flows from “Political Risk” to “Stock Movement” is purely causal.
>
> 2. Another major difference is that we aim to address the bias problem when estimating causal effects (e.g. PEHE, ATE) from text rather than improving the performance (e.g., MAE, MSE) of some specific tasks under the causal lens. More specifically, as mentioned in lines 66-85, previous studies that estimated causal effects from text made the assumption that text only contains confounding covariates. This poses a potential bias because there may be unobserved non-confounding covariates that pertain specifically to either the treatment or the outcome.
>
> 3. We formulated equation (10) to counteract the influence of independent variable $z_o$ when modeling the treatment variable. This step is deemed essential for causal estimation.
>
> 4. As we clarified in our response to R(3.1) the orthogonal loss has not been used in [1]. Furthermore, our approach aligns with [8] and [9], which introduced a masked language model objective in order to adapt text representation, making it more efficient for treatment and outcome prediction.
>
> We will certainly acknowledge and cite [1] in our revised version of the paper, along with a concise discussion outlining the similarities and differences.
>
>
> **R(3.2) Regarding the technical contribution is limited:** As discussed in our response to R(3.1), we tailored equation (10), more specifically, aiming to maximize the probability of  $z_o$ (independent with T) for predicting the treatment. In addition, while the prior works employing VAE and MMD have been explored, a notable contribution of our work is that we identified the causal bias issue arising from inadequate consideration of non-confounding covariates when estimating causal effects (e.g., PEHE, ATE) from text. As we have mentioned in lines 69 -85 and reiterated in our response to R (1.1),  existing approaches suffer from this issue as they are not able to differentiate the non-confounding covariates (e.g., Policy Shifts and Expected Revenue in Figure 1) from confounding ones (e.g., Geographical) under their assumption. To the best of our knowledge, we are pioneers in presenting  an effective solution to address this issue by unveiling interactions between variables through our carefully devised disentanglement model that encourages independence among the variables, ensuring that each variable contributes solely to its designated objective.
>
> **R(4.1) Regarding Baseline quite old:** We would like to point out that some of the experiment results can be viewed as surrogates for more recent baselines. For instance, in the Ablation study, the results for DIVA w/o mlm and DIVA w/o ort shown in Table 2 can be roughly considered as representing a more recent baseline, such as [1]. Nonetheless, we will certainly contemplate adding recent baselines like [1] in the revised version.
>
> **R(4.2) Regarding result (1.096 v.s. 1.003) and the significant results:** In Table 1, the causal estimation results of political risk on stock volatility for the baseline model TextCause, over five runs in terms of square root PEHE, are as follows: 1.079556, 1.102212, 1.121453, 1.099713, and 1.074944. For our model, DIVA, the results over five runs are: 1.000645, 1.00681, 1.000243, 1.007245, and 1.002079. The one-tailed t-test between the two samples is 2.40e-06, and the two-tailed t-test is 4.7908e-06, showing DIVA is significant superiority over TextCase. We will complement the results of the significance test in our revised version.
>
> **R(5) Regarding the more recent related works:** We will certainly mention and cite [2-4] in Related work in our updated version of the paper and include a brief discussion.
>
>
> [5] Pearl, Judea. “On a Class of Bias-Amplifying Variables that Endanger Effect Estimates.” ArXiv abs/1203.3503 (2010): n. Pag.
>
> [6] Wood-Doughty, Zach et al. “Generating Synthetic Text Data to Evaluate Causal Inference Methods.” ArXiv abs/2102.05638 (2021): n. Pag.
>
> [7] Zhang, Weijia et al. “Treatment effect estimation with disentangled latent factors.” AAAI (2020).
>
> [8] Shi, Claudia et al. “Adapting Neural Networks for the Estimation of Treatment Effects.” NeurIPS (2019).
>
> [9] Pryzant, Reid et al. “Causal Effects of Linguistic Properties.” NAACL (2021)

---

### Official Review · Reviewer_5SCp · 2023-08-04

**Soundness:** 3

**Excitement:**

3: Ambivalent: It has merits (e.g., it reports state-of-the-art results, the idea is nice), but there are key weaknesses (e.g., it describes incremental work), and it can significantly benefit from another round of revision. However, I won't object to accepting it if my co-reviewers champion it.

**Missing References:**

[1] Hassanpour, Negar, and Russell Greiner. "Learning disentangled representations for counterfactual regression." International Conference on Learning Representations. 2019.
[2] Wu, Anpeng, et al. "Learning decomposed representation for counterfactual inference." arXiv preprint arXiv:2006.07040 (2020).


**Paper Topic And Main Contributions:**

This paper investigates the problem of causal inference from text. Different from most existing causal inference methods which only adjust for confounders, this work disentangles different factors by proposing a VAE-based framework DIVA. The proposed method is evaluated on both synthetic data and real-world data with promising results.

**Questions For The Authors:**

1. What is the main technical contribution of the proposed method?
2. How can the proposed method verifying the causal assumptions and the causal relations in real-world text data?

**Reasons To Accept:**

1. The studied problem is important.
2. The presentation is clear and easy to follow.
3. The experimental results are promising.

**Reasons To Reject:**

Some concerns for this paper:

First, disentangling different factors in a causal model with neural networks has been studied in previous causal learning literature (e.g., [1,2]). Besides, the representation balancing from treatment and control group is a well-adopted technique in existing causal studies. Therefore, it is important to clearly justify the novelty of this work.

Second, to evaluate the robustness of the proposed method, it is suggested to include evaluation under different parameter settings.

Third, in text data, verifying the causal assumptions and identifying the causal relations would be particularly difficult. How can the proposed method address this problem?


**Reproducibility:**

4: Could mostly reproduce the results, but there may be some variation because of sample variance or minor variations in their interpretation of the protocol or method.

**Reviewer Confidence:**

4: Quite sure. I tried to check the important points carefully. It's unlikely, though conceivable, that I missed something that should affect my ratings.

---

> ### Author Rebuttal · Authors · 2023-08-29
>
> Thank you for your thoughtful comments. We hope the responses below are helpful in addressing your concerns.
>
>
> **R(1) Regarding clearly justifying the novelty of this work:**  We would like to thank the reviewer for bringing the two interesting works to our attention. There exist notable distinctions between [1,2] and our proposed method:
>
> 1. While [1,2] aimed to  disentangle different variables for counterfactual regression in structured numeric dataset such as IHDP, Jobs and Twins, a central contribution of our work is the recognition of biased causal estimation inherent in textual data. This bias arises from inadequate consideration of non-confounding covariates and suboptimal modeling of interactions between these variables. As discussed in lines 554-562, exclusively focusing on non-confounding covariates, without the ability to effectively model interactions, fails to  consistently achieve superior performance in textual data (e.g., TEDVAE v.s. CEVAE on the stock movement scenario). However, our proposed model consistently outperforms all compared baseline models across various evaluation metrics and scenarios, which demonstrates the indispensability of imposing constraints and validates the effectiveness of our proposed model DIVA.
>
> 2. Another key difference lies in the formulation of equation (10), which we designed to counteract the influence of the independent variable $z_o$ during the modeling of  the treatment variable. This step has been demonstrated to be essential for casual estimation, aligning with [3]’s work.
>
> 3. Our approach aligns with [4] and [5], which incorporated a Masked Language Model objective in order to adapt text representation, making it more efficient for treatment and outcome prediction in textual data.
>
> 4. Notably, we constructed a novel dataset which serves as a benchmark for causal inference research in the NLP community.
>
> We will certainly acknowledge and cite [1,2] in our revised version and include a discussion highlighting both the similarities and differences between our work and theirs.
>
>
>
>
>
>
> **R(2) Regarding the evaluation under different parameter settings:** The parameter setting in Table 1 is ($\alpha$=1, $\beta$=1, $\gamma$=0.5, $\epsilon$=1); ,We conduct an experiment on different settings of ($\alpha$=1, $\beta$=10, $\gamma$=0.5, $\epsilon$=4) in eq (14) and eq (15), following your suggestion. We have chosen to compare the two strongest baseline CausalBert and TextCause in Table 1 under the stock movement scenario due to the time limit. Thank you for the valuable question, and we will complement such an experiment in our revised version.
>
>
> |            |        Political Risk         |                               |           Sentiment           |                                            |
> |:----------:|:-----------------------------:|:-----------------------------:|:-----------------------------:|:------------------------------------------:|
> |            |     $\sqrt{\text{PEHE}}$      |      $\delta\text{ATE}$       |     $\sqrt{\text{PEHE}}$      |             $\delta\text{ATE}$             |
> | CausalBert |       0.4992$\pm$0.0007       |       0.0400$\pm$0.0133       |       0.4999$\pm$0.0009       |             0.0531$\pm$0.0136              |
> | TextCause  |       0.5306$\pm$0.0051       |       0.0148$\pm$0.0104       |       0.5337$\pm$0.0047       |             0.0286$\pm$0.0195              |
> |    DIVA    | $\textbf{0.4973$\pm$0.0013} $ | $\textbf{0.0072$\pm$0.0057}$  | $\textbf{0.4966$\pm$0.0018}$  |        $\textbf{0.0103$\pm$0.0061}$        |
>
>
> **R(3) Regarding how the proposed method addresses the problem of causal assumption:** Rather than verifying the causal assumption, we aim to adjust for latent covariates for estimating causal effects from observational textual data. Our assumption (SUTVA, Unconfoundedness, and Positivity) aligns with the established norm for identifying causal effects from observational data [6-9]. Within the context of textual data, the sole atypical assumption we need to adopt to ensure the validity of the estimating procedure is to assume that the textual data carries sufficient information to estimate the causal effects which has been justified in [4,5]. Their theorems (e.g., Theorem 3.1 and Theorem 3.2 in [4]) still hold in our assumed causal graph, primarily because all the backdoor paths were blocked when we conditioned on the text (e.g., the earnings call transcript in Figure 1).
>
> We will make this part clear in the revised version.
>
> **Q(1) Regarding the main technical contribution:**  Please refer to our response to R(1).
>
> **Q(2.1) Regarding verifying the causal assumption:** please refer to the response to R(3)
>
> **Q(2.2) Regarding the causal relation in real word-text data:** This question is indeed of great significance. We would like to share our thoughts here. Unfortunately, the “gold/true” causal effects are generally unavailable because of the fundamental problem which we have mentioned in lines 231-240. But what we can do is to establish a connection between treatment and outcome (e.g. as exemplified by equation (14) and (15)) grounded in domain expertise under appropriate assumption, and then design a model to conduct simulation experiments. If the model can achieve good results in these simulation experiments, we can reasonably assume that our designed model can effectively estimate the causal effect when the underlying hypothesis holds in the real-world scenarios.
>
> Moreover, with enhanced expertise, the formulation becomes more accurate, leading to improved model results. This, in turn, increases the likelihood of accurately estimating causal effects between variables in actual scenarios. While we may lack knowledge of the "gold/true" causal effects within real-world scenarios, the potency of the predicted causal effects can still serve as a valuable resource for making well-informed decisions in practical situations.
>
> [3] Pearl, Judea. “On a Class of Bias-Amplifying Variables that Endanger Effect Estimates.” ArXiv abs/1203.3503 (2010): n. Pag.
>
> [4] Shi, Claudia et al. “Adapting Neural Networks for the Estimation of Treatment Effects.” NeurIPS (2019).
>
> [5] Pryzant, Reid et al. “Causal Effects of Linguistic Properties.” NAACL (2021)
>
> [6] Rubin, Donald B.. “[On the Application of Probability Theory to Agricultural Experiments. Essay on Principles. Section 9.] Comment: Neyman (1923) and Causal Inference in Experiments and Observational Studies.” Statistical Science 5 (1990): 472-480.
>
> [7] Rubin, Donald B.. “Estimating causal effects of treatments in randomized and nonrandomized studies.” Journal of Educational Psychology 66 (1974): 688-701.
>
> [8] Pearl, Judea. “Causal inference in statistics: An overview.” Statistics Surveys 3 (2009): 96-146.
>
> [9] Imbens, Guido and Donald B. Rubin. “Causal Inference for Statistics, Social, and Biomedical Sciences: An Introduction.” (2015).

---

### Official Review · Reviewer_EVAH · 2023-08-11

**Soundness:** 4

**Excitement:**

3: Ambivalent: It has merits (e.g., it reports state-of-the-art results, the idea is nice), but there are key weaknesses (e.g., it describes incremental work), and it can significantly benefit from another round of revision. However, I won't object to accepting it if my co-reviewers champion it.

**Paper Topic And Main Contributions:**

The manuscript sets out to reduce bias by examining the interactions among various variables. Its goal was to separate non-confounding covariates during the estimation of causal effects from text. The method ensures that covariates solely influence their intended objectives, promoting variable independence. Notably, the method introduces a constraint to balance representations from both treated and control groups, addressing selection bias. The experiments span two distinct treatment factors across multiple contexts. The model exhibits superior performance when compared to contemporary leading benchmarks.

**Questions For The Authors:**

Besides the above-mentioned questions, I would like to ask the following ones:

1. What would be the largest challenge to further generalize the method to the nonlinear setting?

2. If the hidden variables are actually dependent on the ground-truth data-generating process, why keep them independent and disentangle them?

**Reasons To Accept:**

1. It considered latent non-confounding covariates, which, according to the authors, was ignored in previous works.

2. Experiments support the claimed superiority in various scenarios.

3. The writing is generally clear.

**Reasons To Reject:**

1. More discussion on the assumed causal diagram would be helpful to really make the problem setting more intuitive since it is the foundation of all the following results.

2. Since the existence of both confounding/non-confounding unobserved covariates have been assumed, why not consider some causal discovery methods that could deal with hidden variables to find the causal graph (e.g., FCI, GIN), instead of making it a hypothesis of the hidden data-generating process? Perhaps it could make the whole framework more complete.

**Reproducibility:**

4: Could mostly reproduce the results, but there may be some variation because of sample variance or minor variations in their interpretation of the protocol or method.

**Reviewer Confidence:**

3: Pretty sure, but there's a chance I missed something. Although I have a good feel for this area in general, I did not carefully check the paper's details, e.g., the math, experimental design, or novelty.

---

> ### Author Rebuttal · Authors · 2023-08-29
>
> Thank you for your thoughtful comments. We hope the responses below are helpful in addressing your concerns.
>
> **R(1) Regarding more discussion on the assumed causal diagram:** As mentioned in lines 66-85, our motivation stems from previous studies that estimated causal effects from text. These studies made an assumption that text only contains confounding covariates. This poses a potential bias because there may be unobserved non-confounding covariates that pertain specifically to either the treatment or the outcome. While attempts have been made to address this problem in causal inference involving *structured numeric data*, as seen in prior works [1] (e.g., the TEDVAE baseline we compared in Table 1), our aim revolves around investigating the possibility of mitigating this bias concern by more accurately estimating causal effects from *text*,  considering the presence of two types of non-confounding covariates (e.g., $z_t$ and $z_c$) .
>
> To achieve this, we proposed using VAE for inferring the variables and designed a disentanglement module aimed at differentiating non-confounding covariates from confounding ones. Moreover, we tailored distinct loss functions to effectively model interactions among diverse variables. This approach ensures that each variable contributes primarily to its specific objective, fostering as much  independence as possible. As we have analyzed in lines 554-562, merely considering non-confounding covariates is insufficient without effectively modelling interactions among the variables. For instance, when applied to textual data within the context of stock movement scenarios, TEDVAE struggles to consistently outperform CEVAE. However, our proposed model DIVA consistently surpasses all baseline models across various evaluation metrics and scenarios, which demonstrates the significance of the constraints we introduced and underscores the  effectiveness of our DIVA model.
>
> We appreciate your valuable suggestion and will definitely include this discussion in the revised version to improve clarity.
>
>
> **R(2) Regarding  causal discovery methods:** Textual data presents unique characteristics that make traditional causal inference methods less straightforward to apply. For example, Fast Causal Inference (FCI) has a difficulty in dealing with high-dimensional data. Textual data has a high-dimensional nature, with a vast number of words or features, making it harder to identify causal relationships among many variables. Generalized Independent Noise (GIN) condition [2] was proposed to estimate linear non-Gaussian latent variable causal model. However, textual data might involve complex nonlinear interactions that are difficult to model accurately. Textual data might contain latent variables, such as political risks, sentiments, or unobservable context. These latent factors can influence both the observed text and the outcomes, leading to confounding and challenges in isolating causal effects. For the reasons highlighted above, we proposed our VAE-based model for disentangling latent confounders instead of directly applying traditional causal inference methods for causal structure discovery.
>
> **Q(1) Regarding the challenge of nonlinear setting:** This is an excellent question. We would like to share our thoughts here. Firstly, achieving identifiability of the causal graph requires more stringent assumptions. This might encompass the Markov assumption, causal sufficiency, acylicity, the additive noise assumption [3,4] or the post-nonlinear assumption [5]. However, these assumptions made by specific models could potentially deviate from reality when applied to  real-world application. Secondly, concisely summarizing nonlinear functions (e.g., equation (14) and (15)) is an open problem [6]. In this context, drawing upon domain expertise is crucial to ensure that these equations closely mirror real-world scenarios.
>
> [1] Zhang, Weijia, Lin Liu and Jiuyong Li. “Treatment effect estimation with disentangled latent factors.” AAAI (2020).
>
> [2] Xie, F., Cai, R., Huang, B., Glymour, C., Hao, Z. and Zhang, K., 2020. Generalized independent noise condition for estimating latent variable causal graphs. Advances in neural information processing systems, 33, pp.14891-14902.
>
> [3] Hoyer, Patrik O., Dominik Janzing, Joris M. Mooij, J. Peters and Bernhard Scholkopf. “Nonlinear causal discovery with additive noise models.” NIPS (2008).
>
> [4]Peters, J., Joris M. Mooij, Dominik Janzing and Bernhard Scholkopf. “Causal Discovery with Continuous Additive Noise Models.” arXiv: Machine Learning (2013): n. Pag.
>
> [5] Zhang, Kun and Aapo Hyvärinen. “On the Identifiability of the Post-Nonlinear Causal Model.” Conference on UAI (2009).
>
> [6] Dominik Janzing, David Balduzzi, Moritz Grosse-Wentrup, and Bernhard Schölkopf. ‘Quantifying causal inﬂuences’. In: Ann. Statist. 41.5 (Oct. 2013), pp. 2324–2358.
>
>
> **Q(2) Regarding the reason to disentangle the variables:** The reason why we disentangle these variables is that we aim to accurately model the interactions between the different variables so that we can learn a more accurate $\hat Q$ to predict the outcomes, thereby reducing the bias in causal estimation. And the reasons why we assume the non-confounding variables please refer to the response of R(1).

---

### Meta-Review · Area_Chair_kdmf · 2023-09-20

**Recommendation:** 3

**Metareview:**

Summary: This paper investigates how to estimate causal effects using text data while addressing the issue of selection bias. It proposes a method that disentangles covariates into treatment-related, outcome-related, and confounder categories, ensuring that covariates only impact their relevant parts. Different from most existing causal inference methods which only adjust for confounders, this work disentangles different factors based on a VAE framework. This paper also introduces a new dataset consisting of 115,880 transcripts. The study includes experiments with two treatment factors in various contexts, and the proposed model outperforms other benchmarks.

This paper studied a fundamental and challenging research problem. The paper is generally well-written and easy to follow. The proposed method is evaluated on different benchmarks with promising results. Most of the reviewers agree that this is a solid contribution but the excitement is relatively limited.

---

### Decision · Program_Chairs · 2023-10-07

**Decision:**

Accept-Findings

**Comment:**

Summary: This paper investigates how to estimate causal effects using text data while addressing the issue of selection bias. It proposes a method that disentangles covariates into treatment-related, outcome-related, and confounder categories, ensuring that covariates only impact their relevant parts. Different from most existing causal inference methods which only adjust for confounders, this work disentangles different factors based on a VAE framework. This paper also introduces a new dataset consisting of 115,880 transcripts. The study includes experiments with two treatment factors in various contexts, and the proposed model outperforms other benchmarks.

This paper studied a fundamental and challenging research problem. The paper is generally well-written and easy to follow. The proposed method is evaluated on different benchmarks with promising results. Most of the reviewers agree that this is a solid contribution but the excitement is relatively limited.